# Volume growth in animal cells is cell cycle dependent and shows additive fluctuations

Clotilde Cadart[1,2]*[†], Larisa Venkova[1,2], Matthieu Piel[1,2], Marco Cosentino Lagomarsino[3,4]*

[1]Institut Pierre-Gilles de Gennes, PSL Research University, Paris, France; [2]Institut Curie, PSL Research University, CNRS, Paris, France; [3]FIRC Institute of Molecular Oncology (IFOM), Milan, Italy; [4]Physics Department, University of Milan, and INFN, Milan, Italy

**Abstract** The way proliferating animal cells coordinate the growth of their mass, volume, and other relevant size parameters is a long-standing question in biology. Studies focusing on cell mass have identified patterns of mass growth as a function of time and cell cycle phase, but little is known about volume growth. To address this question, we improved our fluorescence exclusion method of volume measurement (FXm) and obtained 1700 single-cell volume growth trajectories of HeLa cells. We find that, during most of the cell cycle, volume growth is close to exponential and proceeds at a higher rate in S-G2 than in G1. Comparing the data with a mathematical model, we establish that the cell-to-cell variability in volume growth arises from constant-amplitude fluctuations in volume steps rather than fluctuations of the underlying specific growth rate. We hypothesize that such 'additive noise' could emerge from the processes that regulate volume adaptation to biophysical cues, such as tension or osmotic pressure.

*For correspondence:
marco.cosentino-lagomarsino@ifom.eu (MCL);
clotilde.cadart@berkeley.edu (CC)

Present address: [†]Molecular and Cell Biology Department, University of California, Berkeley, Berkeley, United States

Competing interest: The authors declare that no competing interests exist.

## Editor's evaluation

The regulation of cell growth is crucial for our understanding of how cells control their size as well as how they balance cell mass and volume. While recent studies carefully measured single-cell mass trajectories during the cell cycle, revealing complex growth patterns, the volume growth patterns of animal cells are poorly understood. In this interesting study, Cadart et al. now present high-precision measurements of 1700 HeLa cell growth trajectories and offer evidence for the mechanisms that regulate volume growth-rate fluctuations. This is an important demonstration of cell-autonomous volume regulation.

## Introduction

The regulation of animal cell growth is a central question in cell biology (*Cadart et al., 2019*; *Lloyd, 2013*; *Goranov and Amon, 2010*), but our knowledge is limited by the lack of methods to reliably measure cellular growth at the single-cell level. In the last decade, several sophisticated approaches measuring buoyant mass (*Godin et al., 2010*; *Son et al., 2012*), dry mass (*Park et al., 2010*; *Mir et al., 2011*; *Sung et al., 2013*; *Liu et al., 2020*), and volume (*Cadart et al., 2018*; *Zlotek-Zlotkiewicz et al., 2015*) have produced new data revealing unexpected features at several levels. In particular, in contrast to what has been observed in unicellular organisms such as *Schizosaccharomyces pombe* (*Horváth et al., 2013*; *Cooper, 2013*), *Saccharomyces cerevisiae* (*Di Talia et al., 2007*), or *Escherichia coli* (*Wang et al., 2010*), growth patterns of single animal cells in vitro cannot easily be associated

to a simple growth mode, such as mono-exponential, linear, or bilinear. Instead, single animal cells show complex growth patterns that remain poorly understood to date. Note that to avoid ambiguity we hereon call 'growth speed' the time derivative of mass or volume (e.g., $\frac{dV}{dt}$) and 'mass- or volume-specific growth rate,' the growth speed divided by mass or volume (e.g., $\frac{1}{V}\frac{dV}{dt}$).

So far, studies on single animal cell growth have focused on patterns observed at timescales ranging from several hours to a cell cycle. For HeLa cells, growth was reported to couple with cell size, thus contributing to cell size homeostasis (*Cadart et al., 2018*; *Kafri et al., 2013*). These cells were shown to grow, while in G1, at a faster-than-average volume-specific growth rate if they were born smaller than average (*Cadart et al., 2018*). This finding was paralleled by the observation that inhibition of cell cycle progression or growth pathways has antagonistic effects on mass-specific growth rate or cell cycle progression, respectively (*Ginzberg et al., 2018*). A second type of growth pattern was identified in studies measuring cell mass and showed an association between mass-specific growth rate and cell cycle progression or cell age. One study showed that in L1210 cells that undergo polyploidization mass-specific growth rate follows a bell-shaped dependency on mass over the course of each cell cycle, independently of the increasing mass of the cell as ploidy increases (*Mu et al., 2020*). This suggests that the bell-shape pattern of mass-specific growth rate is a function of cell cycle progression, not mass itself. Two other studies reported that the mass growth speed of several cell types displayed periodic oscillations. Although the characteristics of the oscillations identified differ in the two studies, these results show that mass growth follows an oscillatory pattern that depends on time since birth (*Kesavan et al., 2014*) or time until division (*Liu et al., 2020*).

At shorter timescales (1 hr or less), single adherent cells display cell volume (*Cadart et al., 2018*) and cell mass (*Son et al., 2012*; *Liu et al., 2020*) growth trajectories that vary in time and across cells. These fluctuations have not yet been analyzed in detail, and their origin remains poorly understood. Mass or volume changes in a given time interval are the combined consequence of biosynthesis (via transcription and translation) and mechanisms that import or export mass (import of molecules via endocytosis/exocytosis) (*Son et al., 2015b*) or volume (import of water via osmotic balance, hydrostatic pressure, and membrane turnover) (*Cadart et al., 2019*). Variability in growth can thus result from variability in either or both categories of mechanisms. While seminal studies have revealed the origins of 'noise' in transcription in animal cells (*Raj et al., 2006*; *Padovan-Merhar et al., 2015*), the processes leading to noisy mass and volume growth in animal cells still need exploration.

Crucially, with most of the above-cited studies focusing on cell mass, volume growth remains poorly characterized, although it is clear that it follows independent patterns from cell mass throughout the cell cycle (*Zlotek-Zlotkiewicz et al., 2015*; *Miettinen et al., 2019*; *Son et al., 2015a*). Regarding cell volume, open questions remain regarding both the identification of a mean growth mode (e.g., linear or mono-exponential), and on the determination of the fluctuations around this trend. While the study of mean trends is possible with medium-size datasets (hundreds of cells), the study of fluctuations in growth requires much larger datasets (thousands of cells), which remain difficult to generate for animal cells. In a previous study (*Cadart et al., 2018*), we dynamically measured the volume of single adherent animal cells using a fluorescence exclusion technique (FXm) (*Cadart et al., 2017*). Our results indicated that, on average, volume growth speed increases with cell volume, but the data obtained were not sufficient for deeper analysis. Here, we report an improvement of the FXm method that produces higher throughput in volume readouts. We obtained a dataset of around 1700 single-cell volume curves of HeLa cells, combined with the tracking of key cell cycle transitions (birth, G1/S, and mitosis). The data show that volume-specific growth rate depends on both cell cycle phase and cell volume. Our unprecedented statistical resolution also allows us to investigate the variability in volume growth and to show that it arises from constant-amplitude additive fluctuations of growth speed rather than from fluctuations of the specific growth rate.

## Results
### An improved FXm method produces high-throughput dynamic measurements of single-cell volume

To obtain high-throughput measurements of volume growth of single cells, we improved the fluorescence exclusion-based measurement (FXm) of cell volume we previously developed (*Zlotek-Zlotkiewicz et al., 2015*; *Cadart et al., 2017*). Briefly, this method relies on seeding cells in chambers

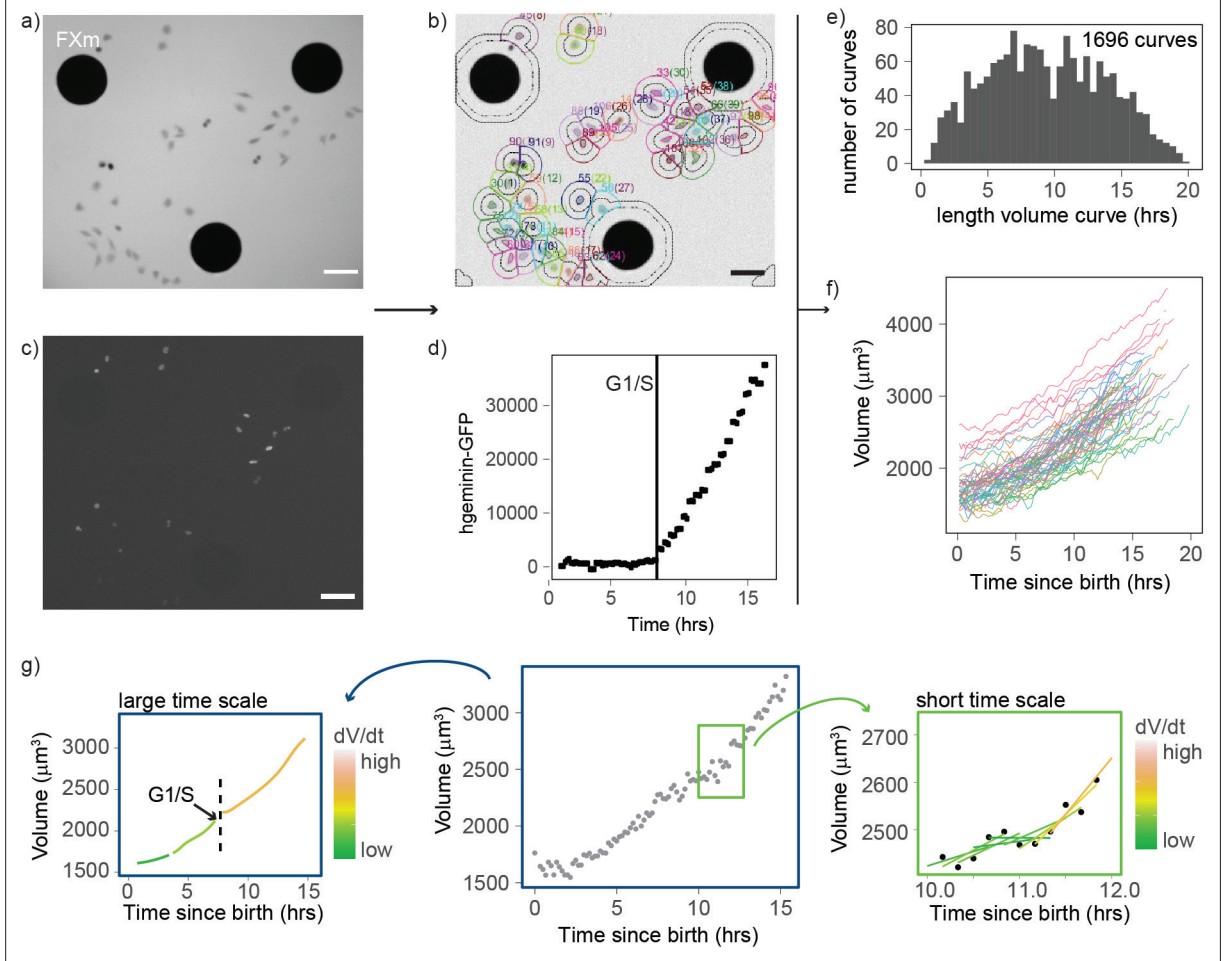

**Figure 1.** An improved fluorescence exclusion-based measurement (FXm) method produces high-throughput dynamic measurements of single-cell volume curves. (**a**) Representative image of a field acquired in an FXm device. The FXm chamber contains a fluorescent dye (Dextran-Alexa-645) that is excluded from the cells and the pillars of the chamber. Hence, the pillars appear in black (large circles), cells are gray, and the background is bright. (**b**) FXm images are automatically segmented using custom MATLAB software, then volume is calculated for each time point. (**c**) Same field as in (**a**) but imaging hgeminin-GFP, a cell cycle marker expressed in the nuclei of the cells. (**a**–**c**: scale bar indicates 100 µm). (**d**) Representative curve of hgeminin-GFP over time in the cell cycle for a single cell. The change of slope in the signal marks the G1/S transition. (**e**) Histogram of the duration of each single-cell volume curve measured. We obtained a total of 1696 curves. (**f**) 72 representative single-cell volume curves from birth to mitosis. (**g**) Representative single-cell volume curve (middle panel) showing trends at timescales of several hours (left panel) and around 1 hr (right panel). Volume growth speed $\left(\frac{dV}{dt}\right)$ was defined from these plots as the time derivative of volume vs. time.

The online version of this article includes the following figure supplement(s) for figure 1:

**Figure supplement 1.** The distributions of growth speed show good agreement between the four experimental replicates.

of known height in the presence of a fluorescent probe (10 kDa dextran) that does not enter or harm the cell (*Figure 1a and b*). Since the cell excludes the dye, the measured fluorescence in the area containing a cell is negatively proportional to the volume of that cell. We combined these volume measurements with cell cycle phase analysis using the hgeminin-GFP part of the FUCCI system (*Sakaue-Sawano et al., 2008*; *Sakaue-Sawano et al., 2013*), the expression of which marks S phase entry (*Figure 1c*). We performed four independent 24-hr-long experiments in which we imaged thousands of cells growing asynchronously in the FXm chambers (*Figure 1—figure supplement 1*). To extract cell volume through time for each individual cell, we developed an automated cell tracking algorithm (*Figure 1b*) and verified that the segmentation and lineage tracing (recording of mitotic events) were accurate by manual inspection of each single-cell trace. Birth was defined as the onset of cytokinesis, the G1/S transition was defined as the onset of increase in hgeminin-GFP intensity (*Figure 1c and d*) and mitosis was defined as the onset of the mitotic volume overshoot (*Cadart et al.,*

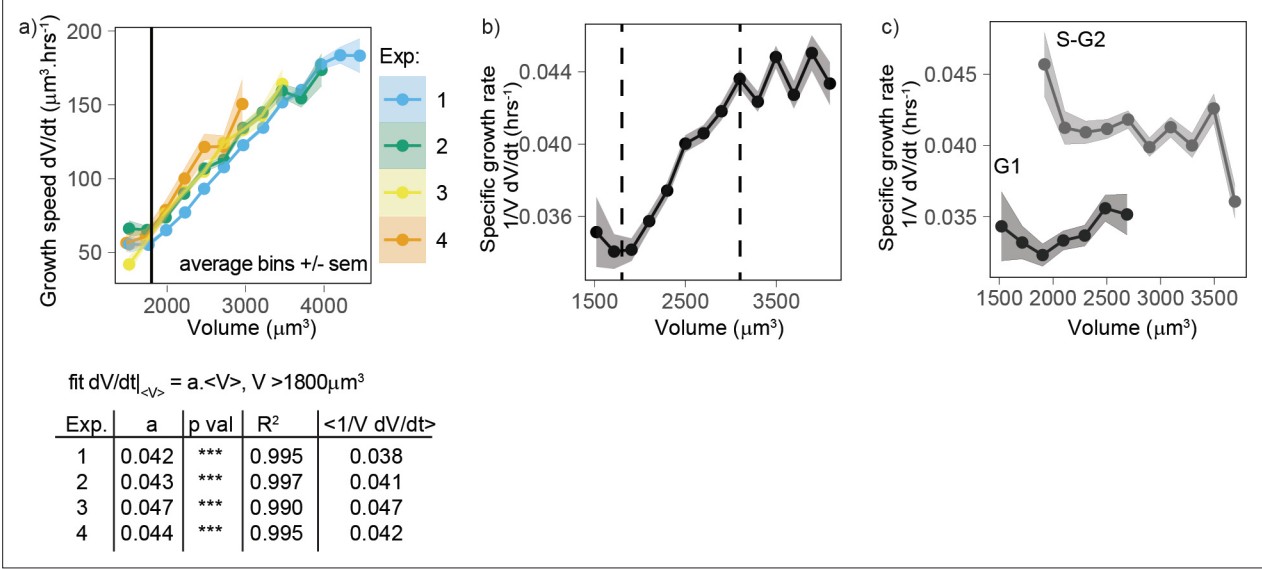

**Figure 2.** Volume growth is close to exponential on average for a wide range of volumes, with a higher specific rate in S-G2 than in G1. (**a**) Top panel: growth speed $\left(\frac{dV}{dt}\right)$ (in units volume/time) as a function of volume for each experiment (points are average bins of width = 250 μm³, the ribbon represents the standard error on the mean for each bin, N = 4, n > 25 different cells in each bin). Bottom panel: table showing the results of the linear fit of growth speed as a function of volume for each experiment. Derivatives are computed on 50 min windows. (**b**) Specific volume growth rate, defined as the binned average of $\frac{1}{V}\frac{dV}{dt}$ at fixed volume $V$, plotted as a function of volume (average bins of width = 200 μm³ ± standard error on the mean). Vertical dashed lines indicate the range where growth rate increases linearly with volume, between 1800 and 3100 μm³. (dots are average bins of width = 200 μm³, the ribbon represents the standard error on the mean for each bin, N = 4, n > 100 different cells per bin). (**c**) Same as (**b**) but grouped by cell cycle stage (G1 vs. S-G2).

2018; *Zlotek-Zlotkiewicz et al., 2015*; *Son et al., 2015a*). Using this approach, we obtained 1696 verified single-cell volume trajectories that contained all available cell cycle information (time of birth, time of G1/S, and/or time of entry into mitosis, *Figure 1e and f*). This high-quality, high-throughput measurement of animal cell volume was used to analyze the patterns and regulation of cell volume growth with unprecedented statistical resolution.

## Volume growth is close to exponential for a wide range of volumes

First, we considered the growth mode of cells – a central question to understanding both cell growth and size homeostasis (*Schmoller, 2017*; *Zatulovskiy and Skotheim, 2020*). Typical limit cases are linear or exponential growth. As previously reported for adherent cell types (*Liu et al., 2020*; *Cadart et al., 2018*), single-cell trajectories show highly variable behavior (*Figure 1g*), making it difficult to associate them with any simple behavior. We turned to an alternative method based on population averages. We reasoned that, in the case of an average simple exponential growth model, growth speed should increase, on average, linearly with volume and the slope $\alpha$ followed by $\frac{dV}{dt}$ vs. $V$ (formally the trend of a conditional average) can be used to define an average volume-specific growth rate (*Cadart et al., 2019*). All four experiments consistently showed that, for volumes higher than 1800 μm³ (and lower than 4000 μm³), growth speed increases on average linearly with volume, with a slope $\alpha$ that was very close to the (unconditional) average of $\langle 1/V\,dV/dt \rangle$ (*Figure 2a*). The agreement of these two different estimates supports the idea that average exponential growth describes these data well. All four experiments were also very similar with values of $\alpha$ ranging from 0.038 to 0.047 h⁻¹. Thus, we conclude that volume growth is faster than linear, and on average close to exponential in this range of cell sizes. Cells with volumes below 1800 μm³ did not follow the same trend, likely due to a different pattern of growth early in the cell cycle (see below).

## Specific volume growth rate depends on cell cycle progression

When we considered in more detail an estimated volume-specific growth rate, defined by the conditional average of $\frac{1}{V}\frac{dV}{dt}$ vs. volume $V$ (which has units 1/time), as a function of volume, we observed a

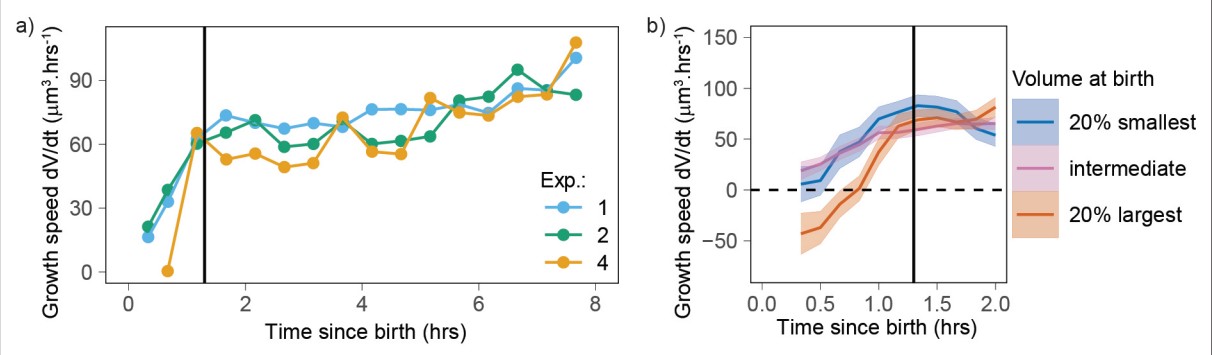

**Figure 3.** Newborn cells show a different trend in growth speed. (**a**) Growth speed (proxied by the discrete derivative $\frac{dV}{dt}$ taken on 50 min windows) as a function of time from birth (onset of cytokinesis) for the three experiments that yielded more than 80 different cells. The vertical line indicates 1.3 hr after birth. Circles are averages bins containing at least 100 cells (N = 3). (**b**) Zoom of the first 2 hr of the data shown in panel a with the data pooled by volume at birth (blue, orange, and pink lines correspond, respectively, to sliding averages of the largest 20% birth volume, smallest 20% birth volume, and the rest of the cells, the dashed horizontal line indicates a growth speed equal to 0, the black vertical line indicates 1.3 hr from birth).

The online version of this article includes the following figure supplement(s) for figure 3:

**Figure supplement 1.** Growth speed as a function of time from birth shows time periodicity for the largest and smallest cells at birth.

slight but significant increase with volume for cells between 1800 and 3100 µm³ (**Figure 2b**). Outside this range, the trend is more complex but the robustness of the observed behavior may be limited due to the lower number of observations at these extreme sizes. We hypothesized that one potential cause of increase in volume-specific growth rate during the cell cycle could be a cell cycle stage dependency, which we could test using data on the transition from G1 to S-G2 phase. To address this question, we repeated the plot of volume-specific growth rate as a function of volume, also grouping the data by cell cycle phase (G1 vs. S-G2). This analysis shows that estimated volume-specific growth rate is nearly constant with volume for a given phase, and growth rate in S-G2 is about 15% higher than in G1 (**Figure 2c**). These results show that progression from G1 into S/G2 is accompanied by an increase in the volume-specific growth rate.

## Newborn cells show a distinct pattern of volume growth

The observation that small cells show a different growth behavior than the rest of the cells (**Figure 2a and b**) prompted us to test whether growing cells could follow different patterns early in the cell cycle. We examined volume growth speed $\frac{dV}{dt}$ as a function of time from birth for the three experiments that had more than 80 cells (to ensure that we had enough statistical power, even when analyzing each experiment separately). In all three experiments, volume growth speed showed a fast increase during the first 1.3 hr after birth and increased more slowly and steadily after that point in time (**Figure 3a**). This suggests that during the initial 1.3 hr after birth, cells follow patterns different from those observed during the rest of the cell cycle. To gain more insight into the details of volume growth during this initial period, we pooled the three experiments together and grouped cells by their volume at birth. The largest cells at birth started their cell cycle with a negative growth speed (meaning that they were losing volume) during the first hour after birth. Small, intermediate, and large cells ultimately converged on the same growth speed at 1.3 hr after birth (**Figure 3b**). We note that in our data, birth is defined as the first time point of cytokinesis onset (a process that is then typically completed within 20–30 min; **Cadart et al., 2018**), thus the 1.3 hr period comprises the end of cytokinesis as well as early G1 phase. The fact that volume follows a distinct pattern early in the cell cycle suggests a different mechanism of volume growth regulation as cells re-enter interphase.

## Volume growth rate fluctuations decrease with cell volume

Our analyses so far indicate that, excluding the initial (**Figure 3**) stage of the cell cycle, volume growth is close to exponential for a wide range of volumes (**Figure 2a**), at a rate that changes with cell cycle phase (**Figure 2c**). Next, since little is known about the cell-specific variations around this average behavior, we set out to evaluate the fluctuations, focusing on the range of volumes for which the growth behavior is well characterized (cells between 1800 and 3100 µm³ [**Figure 2b**] and starting

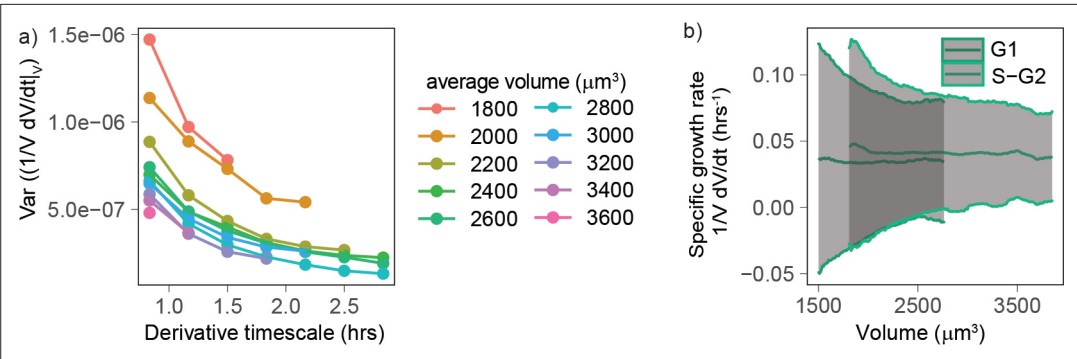

**Figure 4.** Volume growth rate fluctuations decrease with increasing cell volume. (**a**) Variance of specific growth rate quantified by $\frac{1}{V}\frac{dV}{dt}$ using discrete derivatives of fixed volume bins calculated over increasing time windows (x axis) and for groups cells of increasing volume (colored lines). Circles are averages computed for bins that contain at least 100 different cells, N = 4. (**b**) Mean (line) and standard deviation (gray ribbon) of growth rate (quantified by $\frac{1}{V}\frac{dV}{dt}$ using discrete derivatives on fixed volume bins and a timescale of 50 min) plotted as a function of volume and by phase. Values are calculated on sliding windows of 200 µm³, and bins contain at least 100 cells (N = 4).

1.3 hr after birth [**Figure 3a**]). Volume growth is the result of the combination of both biosynthetic pathways that act over the cell cycle and homeostatic pathways (that typically act at shorter timescales) that maintain a balance of cellular osmosis, hydrostatic pressure, and density (**Cadart et al., 2019**; **Neurohr and Amon, 2020**). We first quantified the variability of volume growth by the variance of the specific growth rate, proxied by $\frac{1}{V}\frac{dV}{dt}$, for cells grouped in different volume bins, and we found that this variance decreases rapidly with the timescale $dt$ over which one takes the discrete derivative (**Figure 4a**). Hence, the timescale must be specified for a meaningful comparison of the size of such fluctuations (e.g., across different studies). Moreover, we found that, at fixed timescale of the discrete derivative, the variance in specific growth rate decreases with volume (**Figure 4a**). Since this observation was robust across all derivative timescales, we decided to focus on the fluctuations of growth rate measured at the shortest accessible timescale (50 min, corresponding to five frames). When we plotted together the mean and standard deviation of specific growth rate as a function of volume (**Figure 4b**), the standard deviation of specific growth rate clearly decreased with volume.

## Growth rate fluctuations are dominated by constant noise

To better understand the observation of a reduction in growth rate noise with cell volume, we used a stochastic mathematical model describing cell growth (see Apppendix 1 and **Figure 5a**). This model has a long history of applications outside biology (**Cox, 1997**; **Lo et al., 2011**), but it was recently proposed by **Pirjol et al., 2017** in the context of cell growth. The model considers that cells on average grow exponentially and describes fluctuations around this mean growth as noise:

$$\frac{dV}{dt} = \alpha V + \nu_\alpha V^\gamma$$

where $\alpha$ is the mean specific growth rate and $\nu_\alpha$ is a white noise term. Thanks to the factor $V^\gamma$, this equation interpolates the limit cases of 'additive' (constant amplitude) fluctuations and 'multiplicative' (volume-specific) fluctuations (**Figure 5a**, see also Appendix 1 for details and **Figure 5—figure supplement 1a** for the validation with simulations). Central to the model is the definition of the parameter $\gamma$ ($0 \leq \gamma \leq 1$) that sets the relative weight between two kinds of noise: when $\gamma = 1$, we obtain $dV/dt = (\alpha + \nu_\alpha) V$, and the model describes multiplicative specific growth rate fluctuations (**Figure 5b**). These fluctuations are symmetric with respect to a reference specific growth rate $\alpha$, hence they can be interpreted as emerging from noise in biosynthetic rates (e.g., surface synthesis, protein synthesis, etc.). When $\gamma = 0$, the model describes an additive noise of constant amplitude, acting symmetrically on growth speed $dV/dt$, which can be interpreted as resulting from any of the homeostatic processes that contribute to setting steady-state volume (e.g., homeostatic constraints of biophysical origin on hydrostatic pressure, osmotic pressure, etc.). The model also allows for intermediate values of $\gamma$, which would effectively describe the combined presence of additive and multiplicative noise sources on growth.

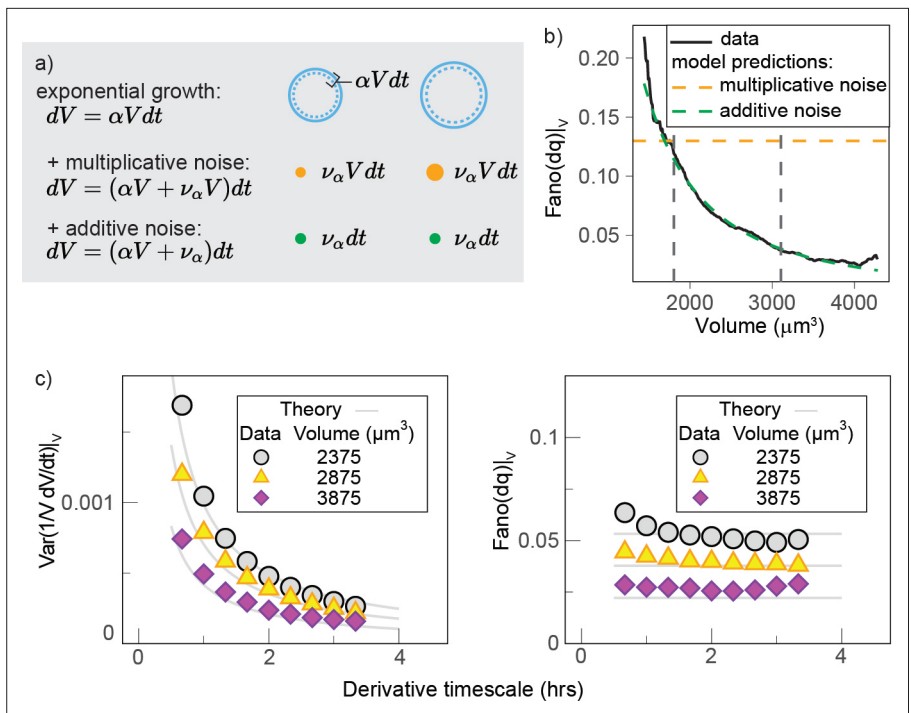

**Figure 5.** Growth rate fluctuations are best described by a model with additive noise. (**a**) Schematic representing the exponential growth of a cell (dashed blue circle) adding volume $dV$ (solid blue circle) proportionately to its volume at a rate $\alpha$ and to timescale $dt$. Two limit cases for fluctuations around this baseline exponential growth are a 'multiplicative noise,' which is volume-specific, hence increases with volume, or an 'additive' noise, whose amplitude is constant. (**b**) Mean-normalized variance (Fano factor) of the conditional 'log return' $dq|_V = \left[\log\left(V\left(t+dt\right)\right) - \log\left(V\left(t\right)\right)\right]|_V$, computed in sliding volume bins, and plotted as a function of volume. The dashed lines represent the theoretical prediction of the model in the case of pure multiplicative (orange dashed line) or pure additive noise (green dashed line), assuming that the amplitude of the noise is set by the fluctuations in the smallest volume bin (yielding a constant line with an intercept $b = \text{Fano}(dq|_{V=1800\ \mu m^3})$ for multiplicative noise, and a power law $\sim b/V^2$ with the same condition for additive noise). (**c**) The Fano factor quantifies fluctuations robustly over different timescales. Left panel: comparison of the variance-specific growth rate (quantified by $\frac{1}{V}\frac{dV}{dt}$ using discrete derivatives on fixed volume) as a function of the timescale of the discrete derivatives in the data (symbols) and the theoretical model predictions (solid gray lines) for different volume bins. Right panel: the Fano factor of the conditional log return $dq|_V$ defined above (same symbols) is robust to timescale changes, as predicted by the model.

The online version of this article includes the following figure supplement(s) for figure 5:

**Figure supplement 1.** Validation of the model.

**Figure supplement 2.** Experimental controls shows that cell volume fluctuations have biological origin, with a minimal contribution from technical noise.

**Figure supplement 3.** Autocovariance analysis shows that the timescale of volume fluctuation is around 760 s.

---

An important prediction of the model (**Figure 5c**) is that the mean-normalized variance of growth rate (Fano factor) is not dependent on the timescale of the derivative. When we examined our experimental data, we observed that this was indeed the case (**Figure 5c**). The Fano factor of growth rate depends on $\gamma$, the mean specific growth rate $\alpha$, and constant $\sigma$ as follows:

$$\text{Fano}\left(dq\right)|_V = \frac{\sigma^2 V^{2(\gamma-1)}}{\alpha_q\left(V\right)}$$

where $\alpha_q$ is the average log-return rate at fixed volume (see Appendix 1), and $dq|_V = [\log(V(t+dt)) - \log(V)]|_V$ is a conditional 'log return' and quantifies the volume-specific growth rate in the presence of possible multiplicative fluctuations. Importantly, other estimators of variability such as variance, SD, or CV do depend on the timescale of the discrete derivatives in both the model and the data. We thus decided to use the Fano factor to estimate the variability in

growth rate. We found that the Fano factor of volume growth rate (measured using the 'log return' $dq$) decreased rapidly with volume and that the trends measured in G1 and S-G2 overlapped well (*Figure 5—figure supplement 1b*). This suggests that the decay in volume growth rate variability is independent of cell cycle phase. We thus pooled the two phases together and compared the decay with our model predictions (*Figure 5b*). Comparison of the model predictions for the two extreme cases ($\gamma = 1$ and $\gamma = 0$) and the data shows that the data are very close to a scenario where $\gamma \simeq 0$ and fluctuations are almost entirely additive (*Figure 5b*).

To estimate whether such additive noise came from extrinsic experimental noise, we conducted a series of experimental controls. We first observe that, to measure cell volume, we use areas much larger than the cell, thus naturally preventing from errors induced by poor segmentation. Importantly, previous validation of the FXm (*Zlotek-Zlotkiewicz et al., 2015*; *Cadart et al., 2017*) as well as new controls (shown in *Figure 5—figure supplement 2a–c*) show that FXm accuracy is independent of the excess area around the cell used to calculate volume. A recent study also showed that volume measurement using FXm is not altered by cell shape changes (*Venkova et al., 2021*). As a second test, we compared the fluctuations of detrended cell volume curves to background areas (*Figure 5—figure supplement 2d and e*). This analysis shows that cell volume fluctuations are about 20-fold larger than those of the background (*Figure 5—figure supplement 2f*). Hence, technical sources of noise (which equally impact background and cell volume measurements) contribute minimally to measured volume fluctuations. Third, we measured volume fluctuations on cells imaged at high temporal resolution (every 30 ms) for time spans of 9 s. At this timescale, cells showed much smaller fluctuations compared with cells measured every 10 min (approximately sixfold less), thus strongly suggesting that the volume fluctuations we report are of biological origin and only appear at timescales larger than 9 s (*Figure 5—figure supplement 2f*). To identify the timescale of such fluctuations, we performed 20 s time-lapse experiments over 1 hr and subtracted a linear trend from the data to compensate for the average growth. The autocovariance function of these data shows that fluctuations decay over a timescale of ~10 min (*Figure 5—figure supplement 3a and b*), and this plot matches the analysis of the 10 min time-lapse experiments (*Figure 5—figure supplement 3c*). Overall, this analysis shows that volume fluctuations come principally from a constant-amplitude noise in instantaneously added volume, rather than from random variations of the volume-specific growth rate itself.

## Discussion

Most recent studies on animal cell growth have considered the patterns of dry or buoyant mass (*Liu et al., 2020*; *Ghenim et al., 2021*). Here, we complement this knowledge by providing a high-throughput dataset of volume trajectories in adhering cells, allowing us to compare the behavior of these different parameters.

During a period following birth that lasts on average about 1.3 hr, volume growth displays a pattern where cells of intermediate and small volume grow slowly while larger cells at birth lose volume. This is very different from the mass pattern previously reported (*Miettinen et al., 2019*). Our volume curves exclude the period in mitosis where cell volume transiently increases by 10–30% in volume called the mitotic volume overshoot (*Zlotek-Zlotkiewicz et al., 2015*; *Son et al., 2015a*). This post-birth period also spans both the period of completion of cytokinesis and the beginning of G1 phase, ruling out the hypothesis that mitosis alone determines the early volume growth pattern. Several studies recently showed that the rate of cell spreading associates with cell volume loss because of the coupled mechano-osmotic regulation of cell volume (*Venkova et al., 2021*; *Guo et al., 2017*; *Perez Gonzalez et al., 2018*; *Adar and Safran, 2020*; *Xie et al., 2018*). We speculate that this phenomenon likely contributes to the decrease in volume as large cells spread following mitotic cell rounding (*Cadart et al., 2014*; *Lancaster and Baum, 2014*) and division. Additionally, while mass growth is known to pause briefly during mitosis, from anaphase to late cytokinesis (*Miettinen et al., 2019*), cell volume undergoes a drastic reversible increase during the same period, causing a drop in cellular density (*Zlotek-Zlotkiewicz et al., 2015*; *Son et al., 2015a*). While mass growth seems to start rapidly after cytokinesis (*Miettinen et al., 2019*), ours and previous observations (*Tzur et al., 2009*) suggest that volume growth is initially slower. The post-birth period we identify therefore points to an interesting period of the cell cycle where regulatory mechanisms related to mechanical tension (cell spreading), osmotic pressure (volume overshoot recovery), and density (balance between mass and

volume increase) compete until volume-specific growth rate reaches a steady behavior that lasts for the rest of the interphase.

Throughout the cell cycle, mass growth has been reported to oscillate periodically in HeLa cells (*Liu et al., 2020*; *Ghenim et al., 2021*). We find that this phenomenon is not simply reflected by cell volume. We found some oscillations in volume growth speed as a function of time from birth only in birth volume outliers, and not matching the period and amplitude of the mass oscillations reported previously (*Liu et al., 2020*; *Ghenim et al., 2021*; *Figure 3—figure supplement 1*). This finding suggests that dry mass biosynthesis and volume growth, while being interrelated, can be independent in specific phases of the cell cycle. It is commonly assumed that volume follows mass, due, for example, to osmotic pressure changes (*Cadart et al., 2019*; *Koivusalo et al., 2009*; *Hoffmann et al., 2009*). Under this assumption, an average exponential growth in volume is explained by an exponential growth in mass. Investigation of the molecular mechanisms coupling mass and volume growth in animal cells is still incomplete. There is consensus on a role of mTORC1 (*Demian et al., 2019*), mTORC2 (*Eltschinger and Loewith, 2016*; *Berchtold et al., 2012*), and of the Hippo pathway (*Perez-Gonzalez et al., 2019*), and one study showed that the YAP/TAZ/Hippo pathway may also regulate cell volume independently of mTORC and protein synthesis (*Perez-Gonzalez et al., 2019*), suggesting that a decoupling can occur at the regulatory level. Of note, time variability in ribosome levels and autocatalysis would lead to a fluctuating mass-specific biosynthesis rate, hence to multiplicative fluctuations (not additive) in mass growth rate. Thus, under the hypothesis that volume strictly follows mass, volume would also likely exhibit multiplicative fluctuations. Studies combining measurements of cell shape, mass, and volume at high time resolution will be particularly important to clarify the complex interplay between these parameters.

We find that volume-specific growth rate is 15% higher in S-G2 than in G1. Mathematical frameworks and experiments clearly showed that growth rate modulation as a function of cell size (*Cadart et al., 2018*; *Kafri et al., 2013*; *Ginzberg et al., 2018*) contributes to cell size homeostasis. The other identified growth modulations along the cell cycle (*Mu et al., 2020*; *Tzur et al., 2009*) and the previously reported mass oscillations (*Liu et al., 2020*; *Ghenim et al., 2021*) do not appear to be cell size dependent and are thus unlikely to contribute to cell size control. The mechanisms driving such growth variations and their role in cell physiology remain mysterious. The molecular pathways underlying size homeostasis (*Tan et al., 2021*; *Liu et al., 2018*; *Zatulovskiy et al., 2020*) may provide some explanations, but the identification and investigation of novel biosynthetic regulatory mechanisms may also be important. For example, the 15% increase in volume-specific growth rate in S-G2 may be the result of a similar increase in protein biosynthesis, but this would not meet the common expectations, given that both transcript levels (*Padovan-Merhar et al., 2015*; *Swaffer et al., 2021*) and ribosome amounts (*Scott et al., 2010*) scale linearly with cell volume – at least within the physiological range of cell volume (*Neurohr et al., 2019*). Future experiments may determine whether other factors such as DNA copy number, translation rate (*Ingolia, 2014*), or import of nutrients and small molecules (*Son et al., 2015b*) play a role in the observed volume growth rate change in S-G2.

The high number of curves in our study allows us to investigate systematically the fluctuations in volume growth. Our observation that volume growth shows additive fluctuations counters the idea that the volume-specific growth rate itself fluctuates (*Figure 5a and b*). What could be the origin of such additive noise and how can we explain the absence of noise on the rate itself? Our controls show that extrinsic experimental noise only minimally contributes to the additive fluctuations we measure (*Figure 5—figure supplement 2*) and that the decay timescale of the constant-amplitude fluctuations is 5–10 min (*Figure 5—figure supplement 3a*). Cell volume is a physical parameter that results from an equilibrium of cell hydrostatic pressure, osmolarity, and membrane tension at timescales of minutes to hours (*Cadart et al., 2019*) and much less understood mechanisms that maintain cell density (*Neurohr and Amon, 2020*; *Oh et al., 2021*; *Knapp et al., 2019*; *Delarue et al., 2018*) by coupling mass and volume at timescales of several hours. The timescale at which we observe the fluctuations and the additive nature of these fluctuations suggest that the measured volume variations may come from the mechano-osmotic processes involved in volume regulation at the minute timescale. Future experiments perturbing these processes (*Venkova et al., 2021*; *Guo et al., 2017*; *Perez Gonzalez et al., 2018*; *Adar and Safran, 2020*; *Jiang and Sun, 2013*) while tracking single-cell volume at high temporal resolution may help understand why noise for such processes is additive. It is also remarkable that no multiplicative noise on volume-specific growth rate is observed. In a scenario

where volume is unidirectionally coupled to mass, even if mass growth rate is noisy (*Shahrezaei and Marguerat, 2015*), one could potentially obtain an apparent volume growth rate with fluctuations that are only due to the coupling. None of the available studies considering cell mass has addressed the question of whether mass biosynthesis fluctuations are themselves mass specific. Many important findings regarding mass growth were made using the Suspended Mass Resonator (*Son et al., 2012*; *Mu et al., 2020*; *Cermak et al., 2016*; *Godin et al., 2010*). However, because it measures cells in suspension, volume fluctuations associated to fast cell shape changes (such as cell spreading or cell migration) are likely minimal in these conditions. The comparison of the noise of mass and volume growth rate on these cells with that of adherent cells obtained with other measurements methods (*Liu et al., 2020*) could be informative.

Finally, our analysis shows that growth rate variability, quantified by variance, SD, or CV is strongly dependent on the timescale used to evaluate discrete derivatives (*Figures 4a and 5c*). This result poses an important caveat for the quantitative comparison of growth rate fluctuations across different studies as absolute values of growth rate fluctuations evaluated in different ways and on different timescales (or smoothing windows) may strongly differ. In general, single-cell growth studies are currently limited by the development of theoretical tools that could quantify the contribution of the different determinants of growth such as size, time, and cell cycle phase that act at different timescales. These tools, together with experimental approaches that allow the combined measurement of several size parameters (mass, volume) concomitantly, are needed to further elucidate the growth patterns of animal cells.

## Materials and methods

### Key resources table

| Reagent type (species) or resource | Designation | Source or reference | Identifiers | Additional information |
|---|---|---|---|---|
| Cell line (*Homo sapiens*) | HeLa hgeminin-GFP | Gift form Buzz Baum lab | | |
| Chemical compound, drug | DMEM, high glucose, GlutaMAX Supplement | Thermo Fisher | 61965026 | |
| Chemical compound, drug | DMEM, high glucose, no glutamine, no phenol red | Thermo Fisher | 31053044 | |
| Chemical compound, drug | GlutaMAX | Thermo Fisher | 35050061 | |
| Chemical compound, drug | Fetal bovine serum | Biowest | S1810-500 | Use at 10% |
| Chemical compound, drug | Penicillin/streptomycin | Thermo Fisher | 15070063 | Use at 1% final |
| Chemical compound, drug | Dextran, Alexa Fluor 647; 10,000 MW, Anionic, Fixable | Sigma-Aldrich | D22914 | Stock at 10 mg/mL in PBS |
| Chemical compound, drug | Fibronectin | Sigma-Aldrich | F1141-1MG | 50 µg/mL in PBS |
| Software, algorithm | Software for FXm image analysis and volume calculation | Available upon request to the authors | RRID:SCR_001622 | |

### Cell line and cell culture

HeLa cells expressing hgeminin-GFP were a kind gift from Buzz Baum's lab (UCL, London, UK). Cells were cultured in DMEM-GlutaMAX media and imaged in DMEM without phenol red, supplemented with GlutaMAX. Both media were supplemented with 10% FBS and 1% penicillin-streptomycin. Cell lines were tested monthly for mycoplasma contamination using the PCR Mycoplasma Test Kit I/C from PromoCell and always came back negative.

### Volume measurement with FXm

The detailed protocol for FXm was described previously (*Cadart et al., 2017*), and the design is described in *Cadart et al., 2018*. Briefly, measurement chambers were replicated in PDMS (cross-linker:PDMS, 1:10). To prevent leakage of the fluorescent dextran from the chamber, 4 mm high PDMS cubes were stuck on top of the inlets before punching 2 mm diameter holes for every inlet. The chamber was then irreversibly bound to the 35-mm-diameter glass-bottom FluoroDish by plasma treatment, coated with fibronectin 50 (µg/mL) for about 30 min, then rinsed and incubated

in phenol-free media overnight. To ensure that cells were in a similar growth phase when starting an experiment, cells were seeded at constant density ($1.9 \times 10^4 \times cm^{-2}$) 2 days prior to the experiment. The day of the experiment, cells were detached by incubating with EDTA for 15–20 min, recovered, and seeded at intermediate density in the measurement chamber (*Figure 1a*). 4 hr after seeding, the media was replaced with equilibrated media containing 1 mg/mL of 10 kDa Dextran Alexa-645. Imaging started 2–4 hr after changing the media. While imaging, cells were kept at 37°C with 5% $CO_2$ atmosphere. Imaging was performed on an inverted epifluorescence microscope (Ti inverted [Nikon] or DMi8 inverted [Leica]) equipped with an LED excitation source. Images were acquired with a CoolSnap HQ2 camera (Photometrics) or an ORCA-FLASh4.0 camera (Hamamatsu). Images were obtained using a low-magnification (10×), low numerical aperture objective (NA = 0.3, phase) every 10 min (FXm measurement, main dataset), and 30 min (hgeminin-GFP imaging). Images were taken every 20 s for the timescale analysis (*Figure 5—figure supplement 3a and b*) and every 30 ms for the background fluctuation analysis (*Figure 5—figure supplement 2f*).

## Software analysis

To extract cell volume and cell cycle information from the images, we used a custom-made MATLAB software (*Cadart et al., 2017*). The software contained an image analysis algorithm previously optimized (*Cadart et al., 2017*) that performed successive image treatments to normalize the background intensity and correct for background inhomogeneity (e.g., due to an inhomogeneous light source). The algorithm then segmented the pillars and background to calibrate the fluorescence intensity signal using: (i) the average background intensity to calculate $I_{max}$ , (ii) the average intensity under the pillars in the chamber to calculate $I_{min}$, and (iii) the known height of the chamber. The software then segmented and tracked single cells (*Figure 1b*) throughout the duration of the movie. If a cell divided, the event was recorded and the lineage tree for that cell recorded. Finally, the cell volume and hgeminin-GFP intensity were calculated over the segmented area for each cell and each movie frame.

The background normalization algorithm required the user to manually set several parameters. To ensure that these parameters were accurately chosen, a graphical user interface allowed the user to visualize the results of the image treatment given a set value for each parameter and assess its validity. This allowed, after a few trial and errors, setting a set of parameter values that were optimal for each set of movies obtained in the same FXm chamber. There were four steps to set these parameter values. First, to detect the pillars and later calculate $I_{min}$, the user manually set a threshold that segmented the pillars (from 0 to 1 on a normalized image), the user also set a 'distance of influence' around each pillar that consisted in an area larger than the pillar where cell volume would not be calculated to prevent potential artifacts of volume calculation due to a shadow caused by the pillars (see *Cadart et al., 2017*). Second, to estimate the background and later calculate $I_{max}$, the user chose a threshold to detect the cells (from 0 to 1 on a normalized image) and set the parameter called 'noise factor.' Third, using these parameter values, the image treatment algorithm was applied to the image. Fourth, on the resulting renormalized image, the user set the parameter values for cell segmentation and tracking: the average size of the mask and the threshold value to detect the cell, the maximum moving distance for a cell from one image to the next, the radius around each detected cell (to prevent measuring cells that were too close to each other), the minimum cell size and a parameter 'sigma' that represented a threshold allowing splitting of an object into two distinct object (e.g., after cell division or when two cells are near each other).

During the optimization phase of the analysis pipeline, we reran the analysis on the same movies using different parameter values to test the robustness of the volume calculated to variabilities in user-defined parameter values. The volume curves obtained were very similar and indicated that errors in volume measurement due to variability in parameter settings were negligible. Once the algorithm parameters were set, the software processed hundreds of movies in batch mode using parallel processing to increase the speed of processing. The analysis of a set of movies coming from one experiment took 2–3 full days of computer processing. At the end of this step, we obtained hundreds of movies showing the tracking results (*Figure 1b*). Each movie was then visually checked to correct any errors of segmentation.

## Visual assessment and manual curation of the single-cell tracks

Each single-cell trajectory was visually checked. We verified that the segmentation and lineage tracing (recording of mitotic events) were accurate. During this manual curation, we visually assessed and noted, for each cell, (i) potential frames when the volume calculated should be excluded from the analysis due to a segmentation error, and (ii) if the cell divided, the frame when the cell started rounding and the frame when the first evidence of cytokinesis occurred. We also checked for and corrected mistakes in the lineage tracking. For the first experiment analyzed, we also noted any frames where a cell was near another cell (typically right after birth when the two daughter cells are near each other or later when two cells bump into each other). We then checked that the presence of a neighboring cell was not affecting the volume curve in an obvious way. Since it did not, we stopped tracking this information for the subsequent experiments. This visual assessment and manual curation, although time consuming, was essential to our analysis because it increased our confidence that any fluctuation seen on the resulting volume curve was not due to an identifiable artifact. The resulting volume curves and hgeminin-GFP signal were then imported into R. Using a graphical user interface, each curve was plotted and the user manually selected, for each cell, (i) the beginning and the end of the mitotic volume overshoot (*Cadart et al., 2018*; *Zlotek-Zlotkiewicz et al., 2015*; *Son et al., 2015a*) on the volume curve and (ii) the point of increase in hgeminin-GFP intensity indicating G1/S transition (*Figure 1d*).

## Volume curve smoothing and calculation of growth speed

To then get into the analysis of volume growth speed and growth rate fluctuations, we developed a cleaning and algorithm that would filter out the clear outliers and smooth fluctuations that are within the noise of our measurement technique. Several algorithms were tested, each time checking visually the resulting comparison of the raw volume measurement with the smoothed, filtered curve. The final algorithm selected worked in two steps. First, to filter out clear outliers, a histogram of values on sliding windows of 11 frames was established and the fourth quantile ($Q_4$) and interquartile range ($IQR$) of that distribution were calculated. Then, points that were above or below $Q_4 \pm 0.9 * IQR$ were removed from the volume curves. Second, a smoothing algorithm based on centered averages on sliding windows of three frames was applied. To calculate growth speed ($\frac{dV}{dt}$), local robust linear fits of single-cell volume curves as a function of time were performed on sliding windows of five frames (all figures except *Figure 4a* where the fits were performed on increasingly long windows of time). The slope coefficient of the fit corresponds to the instantaneous growth speed. We compared this approach to calculating the discrete time derivative by plotting the resulting local fits from both methods and visually concluded that the robust linear fit method gave a more faithful representation of the curve fluctuations.

## Background fluctuations and autocovariance analysis

For the comparison of fluctuations on cell volume curves vs. background areas (*Figure 5—figure supplement 2d–f*) and for the autocovariance analysis (*Figure 5—figure supplement 3*), volume curves were detrended as follows: (i) for the 10 min time-lapse experiment, curves were detrended using smoothing average over windows of eight frames = 80 min; and (ii) for all other experiments (20 s and 30 m time lapse), which were shorter than 80 min, a linear robust fit was performed.

## Model and simulations

The model and analytical calculations to estimate cell-to-cell variability are presented in Appendix 1. These calculations were compared with simulations based on a discrete-time realization of the Langevin equation defining the model (see *Figure 5—figure supplement 1*).

## Statistical analysis

All figure generation and statistical analysis were performed in R. Packages used were 'ggplot2,' 'gridExtra,' 'tidyr,' 'dplyr,' and 'robustbase'.

## Data availability

The dataset, C simulation code of the model, and MATLAB code for data analysis are available on the repository (*Cadart et al., 2022*).

## Acknowledgements

Part of this work was carried out at the Aspen Center for Theoretical Physics. The experiments were performed at the Institut Pierre-Gilles de Gennes (IPGG) and the Institut Curie. CC would like to thank the imaging platform from the Institut Curie PICT-IBiSA and the IPGG, and James Utterback for advice on the simulations. The authors would like to thank Teemu Miettinen, Ethan Levien, Orso Maria Romano, Ludovico Calabrese, and Gabriele Micali for helpful conversations, and Matthew Swaffer and Helena Cantwell for comments on the manuscript.

## Additional information

### Funding

| Funder | Grant reference number | Author |
| --- | --- | --- |
| Associazione Italiana per la Ricerca sul Cancro | 23258 | Marco Cosentino Lagomarsino |
| Agence Nationale de la Recherche | ANR-19-CE13-0030 | Matthieu Piel |
| Agence Nationale de la Recherche | ANR-10-EQPX-34 | Clotilde Cadart Larisa Venkova Matthieu Piel |
| Agence Nationale de la Recherche | ANR-10-IDEX-0001-02 PSL | Clotilde Cadart Larisa Venkova Matthieu Piel |
| Agence Nationale de la Recherche | ANR-10-LABX-31 | Clotilde Cadart Larisa Venkova Matthieu Piel |

The funders had no role in study design, data collection and interpretation, or the decision to submit the work for publication.

### Author contributions

Clotilde Cadart, Conceptualization, Formal analysis, Investigation, Methodology, Software, Validation, Writing – original draft, Writing – review and editing; Larisa Venkova, Investigation; Matthieu Piel, Conceptualization, Funding acquisition, Resources, Writing – review and editing; Marco Cosentino Lagomarsino, Conceptualization, Formal analysis, Funding acquisition, Investigation, Methodology, Software, Supervision, Validation, Writing – original draft, Writing – review and editing

### Author ORCIDs

Larisa Venkova http://orcid.org/0000-0001-5721-7962
Marco Cosentino Lagomarsino http://orcid.org/0000-0003-0235-0445

### Decision letter and Author response

Decision letter https://doi.org/10.7554/eLife.70816.sa1
Author response https://doi.org/10.7554/eLife.70816.sa2

## Additional files

### Supplementary files
• Transparent reporting form

### Data availability
The dataset, R code for the analysis and C code for the simulations are all available on Dryad.

The following dataset was generated:

| Author(s) | Year | Dataset title | Dataset URL | Database and Identifier |
|---|---|---|---|---|
| Cadart C, Piel M, Cosentino Lagomarsino M | 2022 | High throughput measurement of single HeLa cell volume growth and cell cycle progression using FXm | https://doi.org/10.6078/D12M6C | Dryad Digital Repository, 10.6078/D12M6C |

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

## Appendix 1

## Stochastic model of exponential growth with noise

This appendix describes the stochastic growth model developed to interpret the reduction of noise on growth rate as a function of volume (*Figure 5*). We compared our data with the stochastic growth model of *Pirjol et al., 2017*.

Following this study, we write

$$\frac{dV}{dt} = \alpha V + v_\alpha V^\gamma,$$
$$\gamma < 1.$$

(1)

This model interpolates the regimes of additive noise ($\gamma = 0$) from fluctuations of biophysical nature and corresponds to purely multiplicative noise ($\gamma = 1$), corresponding to constant-amplitude fluctuations of the specific growth rate around its typical value α. The intermediate case γ ∈ [0, 1] effectively describes a combination of biophysical noise and growth rate fluctuations Here, we assume that $v_\alpha$ is a zero-mean delta-correlated Gaussian white noise, that is $\langle v_\alpha(t) \rangle = 0$, and $\langle v_\alpha(t) v_\alpha(t+\tau) \rangle = \sigma^2 \delta(\tau)$.

This model was used to compute analytically different proxies of growth rate fluctuations, which we used to describe the data. We also verified the agreement of our theoretical predictions with direct numerical simulations of *Equation 1*.

For $\gamma < 1$, let us look at the expected rate and the rate of the mean in this case. Averaging *Equation 1*, one has that

$$\left\langle \frac{dV}{dt} \right\rangle = \alpha \langle V(t) \rangle,$$

because the average of the noise is zero. Hence, we always have that

$$\frac{1}{\langle V \rangle} \left\langle \frac{dV}{dt} \right\rangle = \alpha.$$

The growth rate in the model can be estimated as the conditional average of the speed at fixed volume. Indeed, since

$$dV = \alpha V dt + \sigma V^\gamma dW,$$

One immediately has that

$$\frac{1}{V} \left\langle \frac{dV}{dt} \right\rangle |_V = \alpha.$$

Considering the variance, with some straightforward algebra, one can obtain from *Equation 1* the expression

$$\mathrm{Var}\left(\frac{dV}{dt}\right)|_V = \frac{\sigma^2 V^{2\beta}}{dt}.$$

This expression is plotted in *Figure 5c*.

Defining q = log(V), one has that

$$dq = \frac{dq}{dV} dV + \frac{d^2q}{dV^2} dV^2 + ...,$$

hence

$$dq|_V = \left(\alpha - \frac{\sigma^2}{2} V^{2\gamma-2}\right) dt + \sigma V^{\gamma-1} dW := \alpha_q dt + \sigma V^\beta dW,$$

(2)

where we have defined β = γ − 1 and

$$\alpha_q = \alpha_q(V) = \left(\alpha - \frac{\sigma^2}{2} V^{2\beta}\right).$$

From *Equation 2*, we compute with some algebra the Fano factor of dq (variance over mean), conditional to volume, which is not dependent on the timescale dt,

$$\text{Fano}(dq)|_V = \frac{\sigma^2 V^{2\beta}}{\alpha_q(V)}.$$

This expression is plotted in *Figure 5—figure supplement 1a*.

