## [Editor Report]

The regulation of cell growth is crucial for our understanding of how cells control their size as well as how they balance cell mass and volume. While recent studies carefully measured single-cell mass trajectories during the cell cycle, revealing complex growth patterns, the volume growth patterns of animal cells are poorly understood. In this interesting study, Cadart et al. now present high-precision measurements of 1700 HeLa cell growth trajectories and offer evidence for the mechanisms that regulate volume growth-rate fluctuations. This is an important demonstration of cell-autonomous volume regulation.

---

## [Decision Letter]

**Decision letter after peer review:**

Thank you for submitting your article "Volume growth in animal cells is cell cycle dependent and shows additive fluctuations" for consideration by *eLife*. Your article has been reviewed by 2 peer reviewers, and the evaluation has been overseen by a Reviewing Editor and Aleksandra Walczak as the Senior Editor. The following individual involved in review of your submission has agreed to reveal their identity: Ran Kafri (Reviewer #2).

Essential revisions:

1) Please provide evidence that potential sources of experimental noise coming from segmentation, autofluorescence, movement are not the root cause of the observed noise. For example with the experiments suggested by Referee 1 point 1.

2) Please try to obtain the timescale of fluctuation correlations by increasing the temporal resolution.

3) Please further explain the rationale for removing outliers or include them in the statistics.

4) Please discuss the connection of your results with previous literature (*eLife* 2018 PMID: 29889021 and Science 2009 PMID: 19589995) and to previously proposed molecular mechanisms

5) Please discuss the possibility that exponential growth in volume is due to exponential growth in mass. You may include a further test modifying osmotic conditions, but it is not a requirement.

*Reviewer #2 (Recommendations for the authors):*

In "Volume growth in animal cells is cell cycle dependent and shows additive fluctuations", the authors use a volume exclusion-based measurements to quantify single cell trajectories of volume increase in HeLa cells. The study represents one of the most careful measurements on volume regulation in animal cells and presents evidence for feedback mechanisms that slow the growth of larger cells. This is an important demonstration of cell autonomous volume regulation.

From these measurements, they describe two insights:

1) A cell's growth rate is a joint function of both cell size and cell cycle stage.

2) Our observation that volume growth shows additive fluctuations.

While the subject matter of the present study is important, the insights provided are significantly limited because the authors failed to place their findings in the context of previous literature. The authors present what seems to be remarkably accurate single cell growth trajectories. In animal cells, a joint dependency of growth rate on cell size and cell cycle stage has been previously reported (see *eLife* 2018 PMID: 29889021 and Science 2009 PMID: 19589995). In Ginzberg et al., it is reported "Our data revealed that, twice during the cell cycle, growth rates are selectively increased in small cells and reduced in large cells". Nonetheless, these previous studies do not negate the novelty in Cadart et al. While both Cadart and Ginzberg investigate a dependency of cellular growth rate on cell size and cell cycle stage, the two studies are complimentary. This is because, while Ginzberg characterise the growth in cell mass, Cadart characterise the growth in cell volume. The authors should compare the findings from these previous studies with their own and draw conclusions from the similarities and differences. Are the cell cycle stage dependent growth rate similar or different when cell size is quantified as mass or volume? Does the faster growth of smaller cells (the negative correlation of growth rate and cell size) occur in different cell cycle stages when growth is quantified by volume as compared to mass?

Separate from the phenomenological findings, the authors provide little mechanistic insight. In recent years, mechanisms of cell size regulation have gained more molecular grounding. At least two studies have shown that the selective growth of small cells is mediated by a cell size checkpoint in G1 (Dev Cell 2021 PMID: 34022133). Further, the linkage of cell mass and cell volume has been linked to NKCC1 (PMID: 31067471) and mTORC1. While phenotypic findings are not inferior to molecular ones, the authors should not ignore previous mechanistic findings -if only to prevent the misleading notion that their phenotypic observations stand in mechanistic vacuum and molecular mechanisms on growth regulation are non-existent.

As for the authors findings on the additive fluctuations in volume growth rate, the authors do not address the following trivial possibility. When considering cell mass, one could easily imagine why larger cells would grow faster. Larger cell mass can correlate with more ribosomes, for example, which can result in faster rates of protein synthesis. Similarly, cells that are larger in mass can have more ion channels or aquaporins, which can affect how fast cells increase their volume. By contrast, if cells increase their water content and become larger in volume only (as is the case in hypoosmotic conditions), I would not expect a proportional increase in growth rate. I suspect that, while the authors measure volume and not mass, the exponential kinetics that they observe still derive from the dependency of growth rate on cell mass (and not cell volume). It is only because mass and volume correlate that the authors observe exponential kinetics in volume. While my suspicion can prove false, it cannot be assumed false until empirically shown to be so. One path to test this is to dissociate volume and mass (e.g. changes in osmolarity).

The authors report a distinction in the growth of newborn cells, as compared to cells at the later stages of cell cycle. These results are potentially interesting but are at jeopardy of being an artifact of the morphological changes that characterise newborn cells. Shortly after cell division, cells are less flat and occupy a higher volume from the glass surface on which they are cultured. These morphological changes distance the cells from the objective focal plane and can distort the measurement light intensity. The authors should find a way to control for this possible artifact or exclude this data.

---

## [Author Response]

Essential revisions:1) Please provide evidence that potential sources of experimental noise coming from segmentation, autofluorescence, movement are not the root cause of the observed noise. For example with the experiments suggested by Referee 1 point 1.

These important comments from reviewer 1 prompted us to perform a careful evaluation of the following potential sources of noise:

Area of volume segmentation and noise on an object of fixed volume

The new Figure 5 – Supplement 2a-c, discussed in the main text, shows that volume calculation using FXm is independent of the area around the cell used to integrate the fluorescence: when we draw areas of increasing dimensions around the cell (see the example in Figure 5 – Supplement 2a) and measure the cell volume, the measured volume for a given cell remains remarkably constant (Figure 5 – Supplement 2b). For the range of areas experimentally used to calculate cell volume, the fluctuations of measured volume for a given cell are not correlated with the size of the area and remain stable around +/-0.2% (Figure 5 – Supplement 2c).

The new Figure 5 – Supplement 2d-f (discussed in the main text), compares fluctuations of detrended cell volume curves and background areas of equivalent size. We show that cell volume fluctuations are about 20-fold larger than the background, supporting the idea that technical sources of noise such as fluctuations of the lamp intensity contribute minimally to measured cell volume fluctuations (Figure 5 – Supplement 2f).

Finally, we agree with reviewer 1 that it would be instrumental to measure a static object such as beads. We tried measuring beads or PDMS debris but these objects scatter light in a way that compromises the correct measurement of the fluorescence. Instead, we decided to measure cells at very high temporal resolution (30 msec) for a short period of time (9 sec), where one can safely assume that the cell does not grow and is similar to a static object. At this time scale, cells showed very small fluctuations compared to cells measured every 10 minutes in our dataset (~6 folds less), confirming that the larger part of the fluctuations we report are not caused by the FXm but are likely of biological origin (Figure 5 – Supplement 2f).

Segmentation errors

Reviewer 1 and 2 asked whether noise could arise from segmentation errors. Such errors are unlikely in our data for multiple reasons. Firstly, in order to calculate cell volume, we typically integrate fluorescence over areas larger than the cells, and we showed above that the error from the excess area is negligible. Secondly, we inspected visually each curve after the automated segmentation algorithm for events where cells encounter each other or remain attached to each other after birth, and excluded the timepoints where cells are badly segmented (see Author response image 1).

**Author response image 1. sa2fig1:** The masks around the cells used for volume calculation are always larger than the cells.

Shape changes

To answer this question, we kindly refer the reviewer to another series of experiments we have performed in *Venkova et al., biorXiv 2021.* Specifically, the correlation between cell area and volume at steady state is weak (Author response image 2 B, left panel) suggesting that the method of measurement is not dependent on cell spreading area. However, for cells in suspension that start spreading, the speed of spreading impacts cell volume and cells that spread slowly seldom change their volume while cells that spread fast do (Author response image 2 B, right panels). These experiments thus support the idea that the speed of spreading, not the change in cell shape due to spreading causes a change in volume. They also rule out the possibility that simply changing shape might impact the measure of the cell volume, since cells that spread slowly display a constant volume, while they drastically change their shape.

**Author response image 2. sa2fig2:** Figure B. Figure adapted from figures 1H and S1B of *Venkova et al., bioRxiv 2021.* Left: At steady state, volume shows a poor correlation with cell spreading area. For cells that are in suspension and begin spreading, the comparison of the speed of spreading area (middle) and corresponding volume curve (right) for HeLa cells shows that cells that spread fast loose more volume than cells that spread slowly. Volume is measured with the FXm, using a protocol and set up very similar to the one used in this present manuscript.

Autofluorescence

To address reviewer 1’s question regarding the potential contribution of cell autofluorescence we performed additional experiments measuring cells *vs.* background areas without cells in FXm chambers that did not contain any fluorescent dextran, using the same illumination parameters as for a typical FXm experiment. The intensity measured for cells is comparable to background areas and very small (~100 A.U.) compared with values typically measured in presence of Dextran to calculate cell volume (~ 6000 A.U.) (see Author response image 3).

**Author response image 3. sa2fig3:** When imaging cells in the far-red channel with the same illumination power and time as the one used to measure Dextran-Alexa645 intensity for FXm, the fluorescence intensity measured is the same for cells and for the background.

2) Please try to obtain the timescale of fluctuation correlations by increasing the temporal resolution.

In the main text and in the new Figure 5 – Supplement 3, we report the results of newly performed 20 sec timelapse experiments over one hour to investigate the timescale of volume fluctuations. The autocvariance function analysis on the detrended curves shows that fluctuations decay over a few minutes (Figure 5 – Supplement 3a-c), a timescale that matches the analysis of the 10 min timelapse experiments

3) Please further explain the rationale for removing outliers or include them in the statistics.

The IQR-criteria is designed to remove only rare and obvious outliers (i.e. a jump in volume of more than 15% in 1 frame -10 minutes- which arguably cannot happen biologically). Fluctuations of smaller range are kept (Author response image 4). We looked back at the raw data and calculated that the IQR filtering removes a total of 337 measurement points out of 99935 initial points (0.03% of the points).

**Author response image 4. sa2fig4:** Three examples of single cell trajectories with raw volume measurement (red dots) and points removed with the IQR filtering (blue dots). The IQR criteria is very stringent and removes only the very large ‘bumps’ in cell volume measured (2 left plots) while it keeps fluctuations of smaller amplitude (right plot).

4) Please discuss the connection of your results with previous literature (eLife 2018 PMID: 29889021 and Science 2009 PMID: 19589995) and to previously proposed molecular mechanisms

We included the following new paragraph in the Discussion discussing these references.

“We find that volume-specific growth rate is 15% higher in S-G2 than G1. Mathematical frameworks and experiments clearly showed that growth rate modulation as a function of cell size^1–3^ contributes to cell-size homeostasis. Other identified volume modulations along the cell cycle^4,5^ including the reported mass oscillations^6,7^ do not appear to be cell-size dependent and are thus unlikely to contribute to cell size control. The mechanisms driving such growth variations and their role in cell physiology remain mysterious. The molecular pathways underlying size homeostasis^8–10^ may provide some explanations, but the identification and investigation of novel biosynthetic regulatory mechanisms may also be important. For example, the 15% increase in volume-specific growth rate in S-G2 may be the result of a similar increase in protein biosynthesis, but this would not meet the common expectations, given that both transcript levels^11,12^ and ribosome amounts^13^ scale linearly with cell volume – at least within the physiological range of cell volume^14^. Future experiments may determine whether other factors such as DNA copy number, translation rate^15^ or import of nutrients and small molecules^16^ play a role in the observed volume growth-rate change in S-G2.”

5) Please discuss the possibility that exponential growth in volume is due to exponential growth in mass. You may include a further test modifying osmotic conditions, but it is not a requirement.

We believe that the experiments suggested here are beyond the scope of our current work – where mass is not measured. We note that the hypothesis that volume follows mass explains exponential growth for volume, but not additive fluctuations. We included the following paragraph discussing the coupling between volume and mass growth:

“It is commonly assumed that volume follows mass, due e.g. to osmotic pressure changes^17^. Under this assumption, an average exponential growth in volume is explained by an exponential growth in mass. Investigation of the molecular mechanisms coupling mass and volume growth in animal cells is still incomplete. There is consensus on a role of mTORC1^18^, mTORC2^19,20^ and of the Hippo pathway ^21^, and another study showed that the YAP/TAZ/Hippo pathway may regulate cell volume independently of mTORC and protein synthesis^21^, highlighting that a decoupling can occur at the regulatory level. Of note, time-variability in ribosome levels and autocatalysis would lead to a fluctuating mass-specific biosynthesis rate, hence to multiplicative fluctuations (not additive) in mass growth rate. Thus, under the hypothesis that volume strictly follows mass, volume would also likely exhibit multiplicative fluctuations.”

Reviewer #2 (Recommendations for the authors):In "Volume growth in animal cells is cell cycle dependent and shows additive fluctuations", the authors use a volume exclusion-based measurements to quantify single cell trajectories of volume increase in HeLa cells. The study represents one of the most careful measurements on volume regulation in animal cells and presents evidence for feedback mechanisms that slow the growth of larger cells. This is an important demonstration of cell autonomous volume regulation.From these measurements, they describe two insights:1) A cell's growth rate is a joint function of both cell size and cell cycle stage.2) Our observation that volume growth shows additive fluctuations.While the subject matter of the present study is important, the insights provided are significantly limited because the authors failed to place their findings in the context of previous literature. The authors present what seems to be remarkably accurate single cell growth trajectories. In animal cells, a joint dependency of growth rate on cell size and cell cycle stage has been previously reported (see eLife 2018 PMID: 29889021 and Science 2009 PMID: 19589995). In Ginzberg et al., it is reported "Our data revealed that, twice during the cell cycle, growth rates are selectively increased in small cells and reduced in large cells". Nonetheless, these previous studies do not negate the novelty in Cadart et al. While both Cadart and Ginzberg investigate a dependency of cellular growth rate on cell size and cell cycle stage, the two studies are complimentary. This is because, while Ginzberg characterise the growth in cell mass, Cadart characterise the growth in cell volume. The authors should compare the findings from these previous studies with their own and draw conclusions from the similarities and differences. Are the cell cycle stage dependent growth rate similar or different when cell size is quantified as mass or volume? Does the faster growth of smaller cells (the negative correlation of growth rate and cell size) occur in different cell cycle stages when growth is quantified by volume as compared to mass?

We have clarified these points in our revised manuscript. In a previous publication (Cadart et al., 2018), we reported a negative correlation between growth speed in G1 and cell volume at birth and hypothesized that it contributed to size control. In this current work, we do not address cell size control because we did not focus on obtaining complete volume curves, from birth to division but instead performed shorter experiments (24 hours) and acquired many short (~10 hours) volume curves. We believe that the higher volume-specific growth rate we observe in S/G2 compared to G1 is likely not related to cell size control, as it is not cell-size dependent. A further caveat (additional to the fact that Ginzberg and coworkers measure mass and our study focuses on volume) is that this study focuses on volume-specific growth rate, while Ginzberg and coworkers, in their statement quoted above, refer to what we call growth speed.

Separate from the phenomenological findings, the authors provide little mechanistic insight. In recent years, mechanisms of cell size regulation have gained more molecular grounding. At least two studies have shown that the selective growth of small cells is mediated by a cell size checkpoint in G1 (Dev Cell 2021 PMID: 34022133). Further, the linkage of cell mass and cell volume has been linked to NKCC1 (PMID: 31067471) and mTORC1. While phenotypic findings are not inferior to molecular ones, the authors should not ignore previous mechanistic findings -if only to prevent the misleading notion that their phenotypic observations stand in mechanistic vacuum and molecular mechanisms on growth regulation are non-existent.

We thank the referee for mentioning these relevant and interesting studies. We included them in the discussion.

As for the authors findings on the additive fluctuations in volume growth rate, the authors do not address the following trivial possibility. When considering cell mass, one could easily imagine why larger cells would grow faster. Larger cell mass can correlate with more ribosomes, for example, which can result in faster rates of protein synthesis. Similarly, cells that are larger in mass can have more ion channels or aquaporins, which can affect how fast cells increase their volume. By contrast, if cells increase their water content and become larger in volume only (as is the case in hypoosmotic conditions), I would not expect a proportional increase in growth rate. I suspect that, while the authors measure volume and not mass, the exponential kinetics that they observe still derive from the dependency of growth rate on cell mass (and not cell volume). It is only because mass and volume correlate that the authors observe exponential kinetics in volume. While my suspicion can prove false, it cannot be assumed false until empirically shown to be so. One path to test this is to dissociate volume and mass (e.g. changes in osmolarity).The authors report a distinction in the growth of newborn cells, as compared to cells at the later stages of cell cycle. These results are potentially interesting but are at jeopardy of being an artifact of the morphological changes that characterise newborn cells. Shortly after cell division, cells are less flat and occupy a higher volume from the glass surface on which they are cultured. These morphological changes distance the cells from the objective focal plane and can distort the measurement light intensity. The authors should find a way to control for this possible artifact or exclude this data.

This issue is addressed in point 5 of the essential revisions above. We added a discussion of the coupling between mass and volume. We note that while mass-volume coupling could explain a mean exponential growth of volume, in this hypothesis, time-variability in ribosome levels and autocatalysis would lead to a fluctuating mass-specific biosynthesis rate, hence to multiplicative fluctuations (not additive) in mass growth rate. Thus, under the hypothesis that volume strictly follows mass, volume would also likely exhibit multiplicative fluctuations.

We also refer to a related work from some of us (Venkova et al., BiorXiv 2021) which shows that cell volume measurement using FXm is independent of cell shape changes (see also Author response image 2 essential revisions).